# Local Adaptive Image Filtering Based on Recursive Dilation Segmentation

**DOI:** 10.3390/s23135776

**Published:** 2023-06-21

**Authors:** Jialiang Zhang, Chuheng Chen, Kai Chen, Mingye Ju, Dengyin Zhang

**Affiliations:** 1School of Computer Science, Nanjing University of Posts and Telecommunications, Nanjing 210046, China; b20031620@njupt.edu.cn; 2School of Bell Honors, Nanjing University of Posts and Telecommunications, Nanjing 210046, China; 1022072019@njupt.edu.cn (C.C.); q19010312@njupt.edu.cn (K.C.); 3School of Internet of Things, Nanjing University of Posts and Telecommunications, Nanjing 210046, China; zhangdy@njupt.edu.cn

**Keywords:** edge-preserving filtering, guided filtering, image segmentation, multiple integrated information

## Abstract

This paper introduces a simple but effective image filtering method, namely, local adaptive image filtering (LAIF), based on an image segmentation method, i.e., recursive dilation segmentation (RDS). The algorithm is motivated by the observation that for the pixel to be smoothed, only the similar pixels nearby are utilized to obtain the filtering result. Relying on this observation, similar pixels are partitioned by RDS before applying a locally adaptive filter to smooth the image. More specifically, by directly taking the spatial information between adjacent pixels into consideration in a recursive dilation way, RDS is firstly proposed to partition the guided image into several regions, so that the pixels belonging to the same segmentation region share a similar property. Then, guided by the iterative segmented results, the input image can be easily filtered via a local adaptive filtering technique, which smooths each pixel by selectively averaging its local similar pixels. It is worth mentioning that RDS makes full use of multiple integrated information including pixel intensity, hue information, and especially spatial adjacent information, leading to more robust filtering results. In addition, the application of LAIF in the remote sensing field has achieved outstanding results, specifically in areas such as image dehazing, denoising, enhancement, and edge preservation, among others. Experimental results show that the proposed LAIF can be successfully applied to various filtering-based tasks with favorable performance against state-of-the-art methods.

## 1. Introduction

Image filtering is a practical technique for suppressing incongruous noise while preserving the intrinsic structure information in images, which has been widely applied in many computer-vision applications, including image denoising [1,2,3], exposure image fusion [4], single-image dehazing [5], etc. Generally, image filtering methods accomplish filtering tasks by designing a translation-variant from the filtered input images to the output images under the guidance of guided images. More and more efforts on image filtering have been made with many meaningful achievements.

In existing remote sensing image dehazing methods [6,7,8], block artifacts tend to occur quite easily. However, RDS makes full use of multiple integrated information sources including pixel intensity, hue information, and especially spatial adjacent information, leading to more robust filtering results which effectively address the occurrence of block artifacts and provide a better solution to the issue. During the process of handling remote sensing images, it is essential to use our proposed method to smooth or enhance [9] the images while maintaining the critically important edge preservation and enhancement effects.

The existing image filtering methods can be mainly divided into two categories: local methods and global methods.

Local methods: The core idea of local methods [10,11,12,13,14,15,16,17,18,19,20,21,22,23,24,25,26,27,28] is to achieve image smoothing solely within local regions in the image. The most representative is the bilateral filter [10,11], which filters the image by regulating the value of each pixel by taking a weighted average of its neighboring pixels. In [12], a fast approximation of the bilateral filter was proposed by imposing the combined color sparseness prior and local statistics prior to the filtering input images. In [13], an accelerated bilateral filter was developed to efficiently compute filtering results via applying a KD-tree sampling algorithm. To achieve satisfactory filtering results, Gavaskar et al. [14] designed an improved bilateral filter with an adaptive Gaussian range kernel. Apart from the bilateral filter, the guided image filter proposed by He et al. in [15] also has a remarkable filtering capability; it derives the filtering result from the guided image using a local linear model. By considering the structural differences between guided and input images, Ham et al. presented an image filtering method in [16] which uses static and dynamic guidance. Sun et al. developed a weighted guided image filter (WGIF) [17] to achieve better-smoothing performance by incorporating edge-aware weighting operators into the filtering process. Inspired by unsharp masking, Shi et al. presented a simplified guided filter in [18] based on filtering prior from a low-pass filter.

Global methods: Different from the local methods, global methods [29,30,31,32,33,34,35,36,37,38,39,40,41,42,43,44] formulate the filtering tasks as a global optimization problem consisting of a fidelity term and a regulation term. Typically, the image filtering task is accomplished via building a global weighted least square model to suppress the noise in the image, as in [29]. Min et al. proposed an edge-preserving image smoothing filter [30], which solves the optimization problem by designing a series of global linear sub-systems. In [31], the side window filtering technique was proposed to effectively preserve the edge information. Xu et al. present an effective approach [45] for structure–texture image decomposition, leveraging the discriminative patch recurrence to develop a nonlocal transform that can better distinguish and sparsify texture components. Motivated by the rapid development of machine learning theory, many global filtering methods [3,46,47,48,49,50,51,52,53,54,55,56,57] have been designed under the deep learning architecture. For instance, a CNN-based joint filter [46] was proposed by selectively taking advantage of the structure information consistent with both guided and input images. The Self2self [3] introduces a self-supervised learning method for image denoising, which trains a denoising network solely on the input noisy image, using dropout on pairs of Bernoulli-sampled instances of the image. A fully convolutional neural (FCN)-based image filter method was developed in [58], which determined the learnable parameter through end-to-end training. However, these methods suffer from poor deployability because trained parameters for one specific application can hardly be applied to other different categories of smoothing tasks, making it difficult to put them into practical use at the current stage.

In this paper, we observe that the pixel in the filtered result tends to be obtained by averaging the similar pixels nearby. Motivated by this observation, we first propose a simple image segmentation method called recursive dilation segmentation (RDS) to partition the image into several regions with similar pixels inside, which directly respects the intrinsic spatial information between two adjacent pixels. Then, constrained by the segmented results obtained from RDS, a local adaptive filtering technique named local adaptive image filtering (LAIF) is developed to filter the input image by selectively utilizing the local pixels. Benefiting from the effective RDS, the filtering process, i.e., to smooth the flat regions and preserve the structure information, can be easily realized under the guidance of segmented results. Different from previous works, the proposed LAIF makes full use of the integrated information with respect to pixel intensity, hue component, and especially spatial adjacent information, which makes the filtered results more accurate and reasonable.

The paper is organized as follows. The motivation of this work is introduced in Section 2 and LAIF is proposed in Section 3. This is followed by the experimental results of the LAIF and state-of-the-art methods in Section 4. Finally, the conclusions are provided in Section 5.

## 2. Motivation

Generally, the task of image filtering is to design a linear translation-variant from a filtering input image *p* to an output image *q* under the guidance of a guided image *I*, which can be defined as:(1)$$q_{x}=\sum_{y\in \omega }\;W_{\mathrm{xy}}\left(I\right)p_{y}$$
where x and y are pixel indexes, $W_{\mathrm{xy}}\left(I\right)$ is a designed kernel weight to cope with the linear relationship between *p* and *q*, and ω is an N-by-N local patch. This linear translation-variant model is employed in the proposed LAIF and there is no doubt that the design of the kernel weight is extremely important in the translation-variant model.

The inspiration of the design of the kernel weight in this work comes from two two classic edge-preserving filters, i.e., bilateral filtering (BF) [10] and guided image filtering (GIF) [15]. Before introducing the proposed LAIF, it is necessary to review these two classic filtering algorithms.

In bilateral filtering, $W_{\mathrm{xy}}$ depends on the distance between pixels and the pixel intensity differences. Mathematically, it can be formulated as
(2)$$W_{\mathrm{xy}}^{BF}\left(I\right)=\frac{1}{K_{x}}\exp\left(-\frac{{∥x-y∥}^{2}}{\sigma_{d}^{2}}-\frac{∥I_{x}-I_{y}{∥}^{2}}{\sigma_{r}^{2}}\right)$$
where $K_{x}$ is a normalization coefficient, and $\sigma_{d}$ and $\sigma_{r}$ are the standard deviations in the spatial domain and range domain, respectively.

Figure 1a shows the filtering result of BF. Notably, the filtering ability of BF is remarkable, which can be explained as follows: according to Equation (Equation 2), the output pixel is more likely to be influenced by the nearby pixels with similar distance information and the pixel intensity information (see the weight kernel shown in the figure), thus, the pixels in flat regions are smoothed by the similar pixels nearby and the edges are preserved.

Different from BF, GIF is developed based on a common assumption: the filtering output *q* is a linear transform of the guided image *I* in a local window, which can be expressed as
(3)$$q_{x}=a_{k}I_{x}+b_{k},\;\;\forall x\in {\;\;\;\omega \;\;\;}_{k}$$
where ${\;\;\;\omega \;\;\;}_{k}$ is a local window centered at pixel k, and $a_{k}$ and $b_{k}$ are two constant coefficients in ${\;\;\;\omega \;\;\;}_{k}$.

By formulating the filtering task into an optimization problem, the solution is given by
(4)$$a_{k}=\frac{\frac{1}{\left|\omega \right|}\sum_{x\in \omega_{k}}I_{x}p_{x}-\mu_{k}{\bar{p}}_{k}}{\sigma_{k}^{2}+\epsilon }$$
(5)$$b_{k}=p_{k}-a_{k}\mu_{k}$$
where $\mu_{k}$ and $\sigma_{k}^{2}$ are the mean and variance of the guided image in ${\;\;\;\omega \;\;\;}_{k}$, $p_{k}$ is the mean of the input filtering image in $\omega_{k}$, $\left|\cdot \right|$ is an operator returning the number of pixels inside, and ϵ is a regularization parameter used in optimization problem. The kernel weight can be obtained by averaging $a_{k}$ and $b_{k}$ to ${\bar{a}}_{i}$ and ${\bar{b}}_{i}$.

Figure 1b gives the filtering result of GIF. As analyzed in [15], GIF has a strong filtering capability: for pixels in the “low-variance” area, where the pixels have similar intensity information, their value becomes the average of the neighboring pixels, while for pixels in the “high-variance” area, their pixel values are nearly unchanged.

As mentioned above, both BF and GIF smooth the image based on the local pixels with similar properties with respect to distance information or intensity information. Therefore, for the pixel to be smoothed, only the similar pixels nearby need to be utilized to obtain the filtering result. This motivates us to segment the image into several regions, where the pixels in one partition region share a similar property, before locally smoothing the image based on the segmented results.

## 3. Proposed Method

In this section, a simple but effective image filter technique, i.e., LAIF is proposed. Only two modules are included in LAIF, i.e., the recursive dilation segmentation module and the local adaptive image smoothing module.

### 3.1. Recursive Dilation Segmentation Module

Before introducing recursive dilation segmentation (RDS), we first define the dissimilarity between two pixels in the guided image, which is an RGB-channel image I in this work, by considering both pixel intensity distance and hue distance, and mathematically describe it as:(6)$$\delta \left(x,y\right)=∥I_{x}-I_{y}{∥}^{2}+\lambda \cdot {∥H_{x}-H_{y}∥}^{2}$$
where $\delta \left(x,y\right)$ is the quantifying dissimilarity between x and y, *H* is the hue component of RGB image I, and λ is a balance weight parameter that makes a trade-off between two different categories of distance. In this way, two pixels can be defined as ‘similar’ if
(7)$$\delta \left(x,y\right)<\varepsilon$$
where ε is a threshold parameter that provides an estimation of the maximum possible δ between two similar pixels.

Given the criteria for judging whether two pixels are similar, the image can be easily segmented via RDS.

The fundamental idea of RDS is to partition the image into several segmentation regions in which each pixel belonging to the same segmentation part has similar spatial information and pixel intensity, in a recursive dilation way. More specifically, the image will be gradually partitioned into several non-overlapping regions via several segmentation operations. For the *i*th segmentation operation, we firstly select one pixel that has not been traversed in the image as the dilating center pixel, then, all its similar neighboring pixels will be judged according to Equation (Equation 7) before recursively dilating to them and treating them as the new dilating center pixels. The dilation procedure in each segmentation operation will continue recursively until all the adjacent similar pixels have been traversed. All the pixels traversed within the same *i*th segmentation operation will be stored into the same partition pixel set $S^{i}$. This operation will be executed multiple times until all the image’s pixels are traversed. The segmentation process is described in Figure 2 and Algorithm 1.
**Algorithm 1:** Proposed LAIF
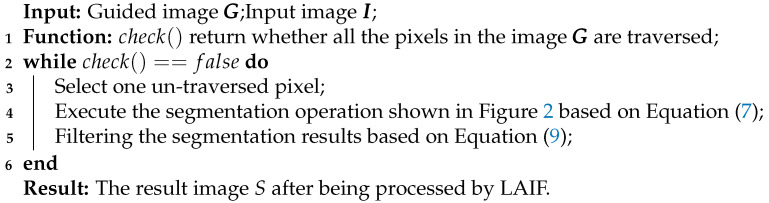


It is worth mentioning that the proposed RDS algorithm is different from the clustering-based image segmentation methods presented in [59,60] because RDS directly applies spatial adjacent information in a gradual extension way. Benefiting from this property, the partitioned pixels in the same pixel set are more analogous with respect to both spatial information and color information.

### 3.2. Local Adaptive Image Smoothing Module

Different from the linear translation-variant model expressed in Equation (Equation 1), in this work we make an assumption that the pixels in the filtering output are only related to the pixels with similar properties within a local window in the filtering input image, which can be formulated as
(8)$$q_{x}=\sum_{y\in S_{x}\cap \omega_{x}}W_{\mathrm{xy}}\left(I\right)\cdot p_{y}$$
where $\omega_{x}$ is a local patch with the size *s* of N×N centered at pixel x, and $S_{x}$ is a partition pixel set that the centering pixel x belongs to. In GIF, the pixels will be smoothed by a local average operation if they are surrounded by similar pixels. Accordingly, a local average operation is employed in LAIF as well, to smooth the target pixels by the similar pixels nearby, thus, the filtering kernel weight can be obtained as
(9)$$W_{\mathrm{xy}}\left(x\right)=\frac{1}{\left|S_{x}\cap \omega_{x}\right|},\;\;y\in S_{x}\cap \omega_{x}$$

The essence of this weight parameter is to selectively utilize local information for local adaptive smoothing. The benefit of doing so is that the pixel information with respect to spatial distance, pixel intensity, and the hue component can be effectively integrated and utilized. Thus, by substituting Equation (Equation 9) into Equation (Equation 8), the filtering result can be obtained as
(10)$$q_{x}=\sum_{y\in S_{x}\cap \omega_{x}}\;\frac{1}{\left|S_{x}\cap \omega_{x}\right|}\cdot p_{y}$$
and the whole algorithm is illustrated in Figure 3.

As shown in Figure 4, LAIF has a remarkable capability to smooth the flat regions and preserve edge information in contour regions. The filtering capability of LAIF can be explained intuitively as follows. Considering the pixels in the flat regions surrounded by many similar pixels, the local average operation will be employed on them, thus, the corresponding regions can be smoothed as shown in Figure 4a. As for the edge regions, two pixels on different sides are bound to belong to different pixel sets because of the distinct color and intensity information. This means pixels across the edge are not averaged together, making conspicuous edges emerge between them, as shown in Figure 4b, so that the edge information is well preserved. However, if we directly apply filtering to the overall segmentation results, the final outcome is quite likely to be influenced by poor segmentation results. Therefore, we have employed the method of iteration between segmentation and filtering. In the joint iterative process of segmentation and filtering, each step of optimization can provide a more accurate input for the next step. In this way, segmentation and filtering can influence and optimize each other in every iteration, resulting in an overall output that comes increasingly close to the optimal solution. This process is illustrated through Algorithm 1.

## 4. Experiments

In this section, a series of experiments in various computer vision tasks are conducted to evaluate the performance of LAIF compared with state-of-the-art methods, i.e., GIF [15], JBF [19], EBF [20], and SDF [16]. The parameters used in the compared methods were adjusted according to the corresponding references. All the experiments were conducted in MATLAB R2020a, on a PC with Intel(R) Core(TM) i7-8750H CPU @ 2.20 GHz (12 CPUs), 2.2 GHz.

### 4.1. Parameter Study

In this section, we conduct an experiment to study two key parameters in this method, i.e., pixel distance threshold ε and local filtering patch size *s*. According to Equation (Equation 7), ε determines the criteria for judging whether two pixels are similar, the image will be partitioned into fewer but larger regions with the increase in ε. As for the local filtering patch size, it determines the number of selected pixels when implementing local adaptive filtering. A larger local filtering patch size makes more pixels with similar properties be averaged together according to Equation (Equation 10). To investigate the effect of ε and the size of the filtering patch *s*, we filter two images under the guidance of themselves using LAIF with different values of ε and different sizes of the filtering patch. The different filtered results are given in Figure 5. As demonstrated from the figure, LAIF with a larger ε tends to show a stronger smoothing ability, effectively removing the unwanted noise in the texture region, while the results via a larger filtering patch size seem more uniform and smooth. Therefore, the filtering ability of LAIF can be changed flexibly by adjusting the values of ε and *s*, enabling LAIF to be applied to different filtering tasks.

### 4.2. Dehazing

The task of image dehazing is to remove haze from a single image. In traditional methods [61], the coarse transmission can be estimated via the patch-wise prior and optimized by an edge-preserving filter to obtain the accurate transmission map. In this section, we consider the hazy image as the guided image and optimize the coarse transmission via different filtering methods. Figure 6 shows the dehazed images restored by the corresponding optimized transmission map. Notably, there exist some halo effects near the edge regions for GIF, JBF, EBF, and SDF (see the highlighted zoomed-in panels), which means that the block effect is not eliminated effectively by the filtering method. On the contrary, LAIF can effectively optimize the transmission map under the guidance of input hazy images. We selected two publicly available datasets, I-HAZE [62] and O-HAZE [63], and used the DCP [61] method to perform haze removal. The results were evaluated using the SSIM and PSNR metrics. The results are shown in Table 1.

For the I-HAZE dataset, our proposed method is among the top performers in both SSIM and PSNR. With an SSIM score of 0.7274, it slightly trails EBF’s 0.7297 but surpasses the performance of GIF, GBF, and SDF. In terms of PSNR, our method outperforms all others, with a score of 28.1024, slightly edging out EBF’s 28.0868. When it comes to the O-HAZE dataset, our method maintains its strong performance. It leads in SSIM scoring, achieving 0.7124, which is marginally better than EBF’s 0.7002. With PSNR, our method again secures one of the top two positions. It scores 28.2481, slightly higher than GBF’s 28.2235. In summary, across both datasets and two distinct metrics our proposed method demonstrates consistently strong performance, often achieving or being very close to the best score among all of the tested methods.

### 4.3. Denoising

In this section, we investigate the denoising performance of LAIF via denoising a no-flash noisy image under the guidance of the corresponding flash image. A comparison between the proposed LAIF and state-of-the-art methods, i.e., GIF, JBF, EBF, and SDF is provided in Figure 7. Visually, the edge texture information of the recovered results via EBF and SDF are prone to blurriness. For the denoising result by JBF, some texture details are removed, which does not correspond to the guided image. In comparison, our recovered results are clean and the structure information is well preserved. We conducted experiments related to image denoising, including the evaluation of flash/no-flash denoising and standard denoising. We selected the FAID [64] dataset for our experiments. These images from the FAID dataset are a subset of the Flash and Ambient Illuminations Dataset. In the evaluation process of flash/no-flash denoising, we added Gaussian noise to the ambient light images and used the flash images as guidance. Additionally, we also employed the standard denoising methods (Gaussian filter denoising [65], median filter denoising, total variation denoising [66]) for direct denoising operations.

Table 2 evaluates various denoising methods using SSIM and PSNR metrics. The proposed LAIF method excels in flash/no-flash denoising, achieving the highest scores (SSIM: 0.7588, PSNR: 28.5654), indicating superior structure preservation and noise reduction. In standard denoising, LAIF remains competitive, surpassing Gaussian filter and total variation in SSIM and nearly matching median filter in PSNR.

### 4.4. Detail Enhancement

The fundamental idea of detail enhancement is to magnify the high-frequency detail layer, i.e., the difference between the input image and the filtering output image. Figure 8 gives the enhanced images obtained by the combination of the boosted detail layer (three times magnification) and the input image. As shown in the figure, JBF and SDF tend to introduce a gradient reversal effect into the results. In comparison, the enhanced results via the proposed LAIF can highlight the detail information in the images without producing negative visual effects, e.g., halo artifacts or gradient reversal. In the evaluation of detail enhancement, we selected the publicly available LIVE Release2 [67] dataset. This dataset contains various types of distorted images. We enhanced the details of images distorted by JPEG, JPEG2000, and Gaussian blur, and then evaluated them using the SSIM and PNSR metrics. The results are shown in Table 3. Our proposed method generally exhibits superior performance in the SSIM metric across all datasets. This suggests that our method is highly effective in enhancing structural details in images while preserving their natural appearance. In conclusion, these results overall suggest that our method consistently delivers strong performance in detail enhancement.

### 4.5. Edge Preserving

Edge preserving is a necessary ability for ‘good’ image filtering. We first obtain segmented guidance images and segmentation information through recursive dilation. These are then fed into the filter to perform filtering operations on the image. The filtered result represents the edge-preserved outcome. For other methods, we used the official smoothing techniques provided to assess their edge-preservation effects. To evaluate the edge-preserving ability of the proposed LAIF, we provide the filtering results of state-of-the-art filtering methods, and the corresponding one-dimensional illustrations, in Figure 9. It can be observed from the figure that GIF, JBF, and EBF are prone to streak artifacts, deviating the edges from their original shapes. In contrast, the filtered results via the proposed LAIF are free of halo effects and the edge information is well preserved. Moreover, LAIF can smooth the noise in flat regions, as shown in the result. Because the effect of edge preservation cannot be directly quantified and the effect of edge preservation is often reflected in its dehazing ability, we can evaluate the edge preservation and dehazing ability together. This is because in traditional dehazing methods, the better the smoothness and edge preservation when estimating the transmittance, the better the dehazing effect. Therefore, the effect of edge preservation is also reflected in its dehazing ability. The results are shown in Table 1.

## 5. Conclusions

In this paper, we observe that the pixel in a filtered result can be obtained using similar pixels nearby. By applying the recursive dilation segmentation method, an image can be partitioned into several segmentation regions, the pixels within each of which share a similar property. The main advantage of this method is that it can directly make use of the intrinsic spatial adjacent information between two neighboring pixels in a recursive dilation manner. Moreover, the pixel intensity information and hue component information are also considered in the dilation process. Benefiting from this image segmentation method, the unwanted texture details in the flat regions can be effectively smoothed while the structure information in the edge regions can be well preserved, via employing a local adaptive image filter derived from RDS. In addition, the application of LAIF in the remote sensing field has achieved outstanding results, specifically in areas such as image dehazing, denoising, enhancement, and edge preserving, among others. We investigate the filtering ability of LAIF by conducting a series of experiments between LAIF and the state-of-the-art methods, demonstrating that it performs favorably against the others in terms of both filtering quality and generic ability.

## Figures and Tables

**Figure 1 sensors-23-05776-f001:**
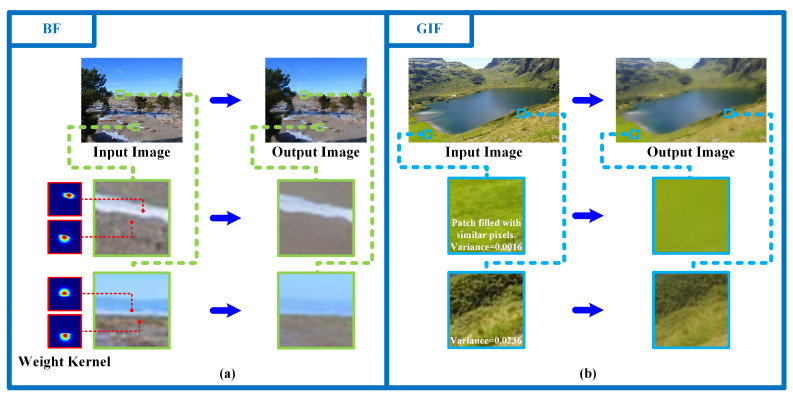
Demonstration of filtering processes of BF and GIF. (**a**) The filtering process of BF and an enlarged section of the result. The red frames show the filtering weight kernel of two pixels in the image (red means a larger weight and blue means a smaller weight). (**b**) The filtering process of GIF and an enlarged section of the result. The white words in the zoomed-in panel are the description of the panel.

**Figure 2 sensors-23-05776-f002:**
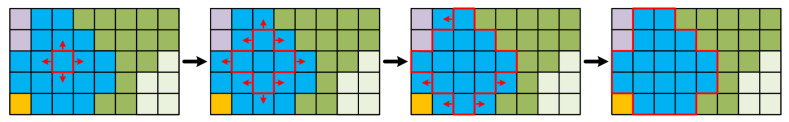
Illustration of one segmentation operation. The squares with the same color denote pixels with similar properties based on Equation (Equation 7), the red frame is the traversed region, and the red arrows stand for the dilation directions.

**Figure 3 sensors-23-05776-f003:**
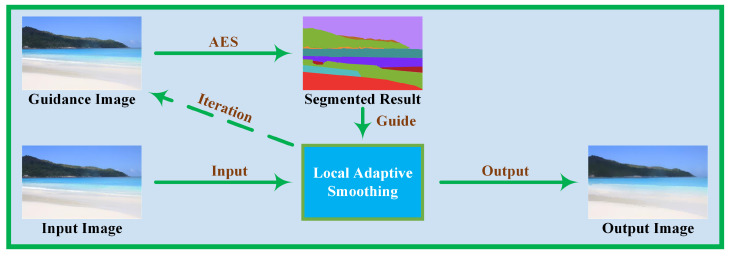
Illustration of the proposed LAIF.

**Figure 4 sensors-23-05776-f004:**
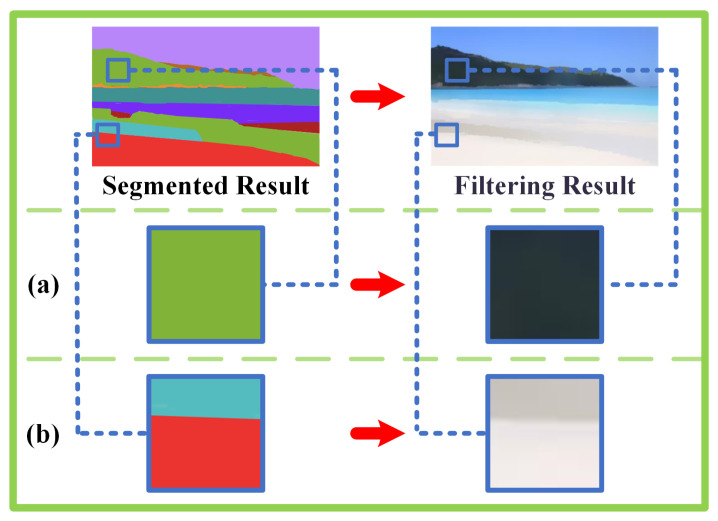
Illustration of the segmented result and the corresponding filtering result of LAIF. The (**a**) is the interior of the segmentation result and The (**b**) is The boundary of the segmentation result.

**Figure 5 sensors-23-05776-f005:**
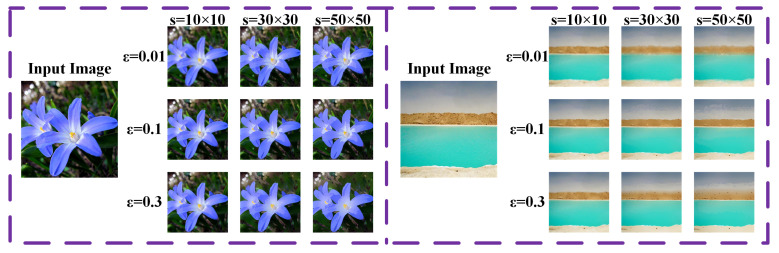
The results of LAIF using different values of ε and sizes of filtering patch *s*.

**Figure 6 sensors-23-05776-f006:**
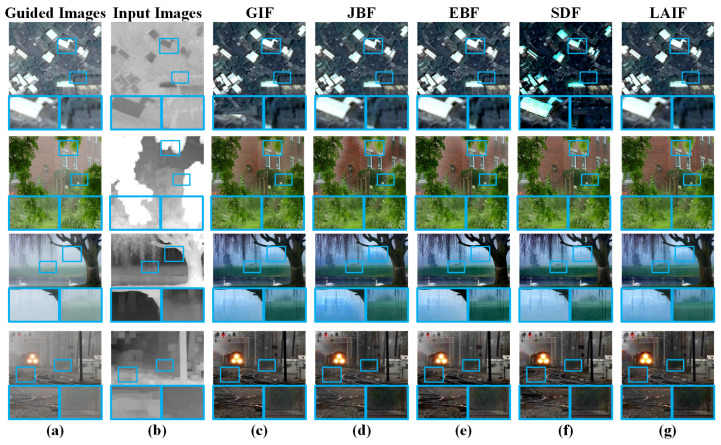
Dehazing results obtained by the optimized transmission map via LAIF and other state-of-the-art methods, and the corresponding zoomed-in patches in the blue frames. (**a**) Hazy images as the guided images. (**b**) Coarse transmission map as the input images. (**c**) GIF. (**d**) JBF. (**e**) EBF. (**f**) SDF. (**g**) LAIF.

**Figure 7 sensors-23-05776-f007:**
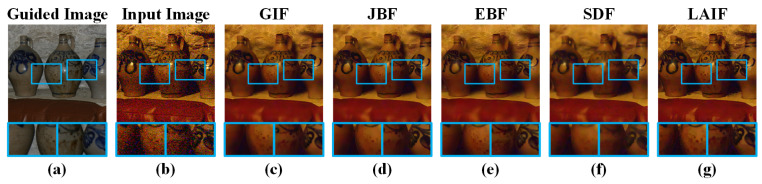
Filtering results via LAIF and other state-of-the-art methods, and the corresponding zoomed-in patches in the blue frames. (**a**) Flash images as guided images. (**b**) No-flash images as input images. (**c**) GIF. (**d**) JBF. (**e**) EBF. (**f**) SDF. (**g**) LAIF.

**Figure 8 sensors-23-05776-f008:**
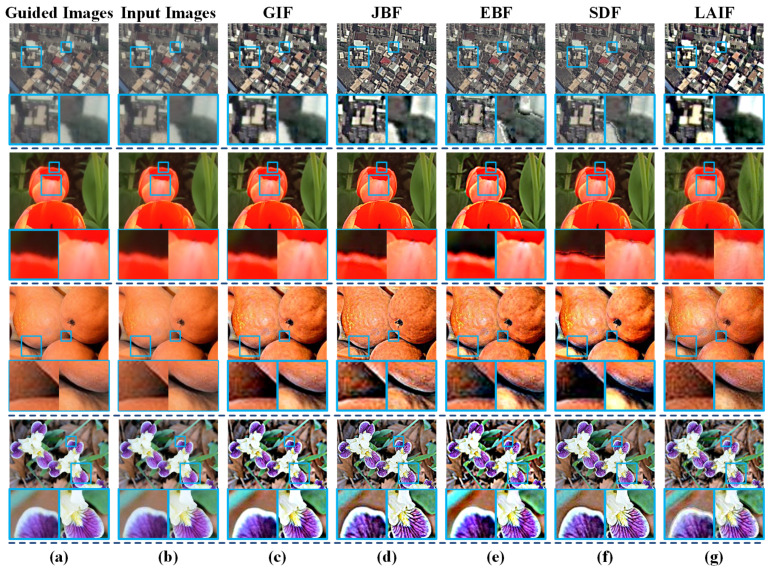
Detail enhancement results via LAIF and other state-of-the-art methods, and the corresponding zoomed-in patches in the blue frames. (**a**) Guided images. (**b**) Input images. (**c**) GIF. (**d**) JBF. (**e**) EBF. (**f**) SDF. (**g**) LAIF.

**Figure 9 sensors-23-05776-f009:**
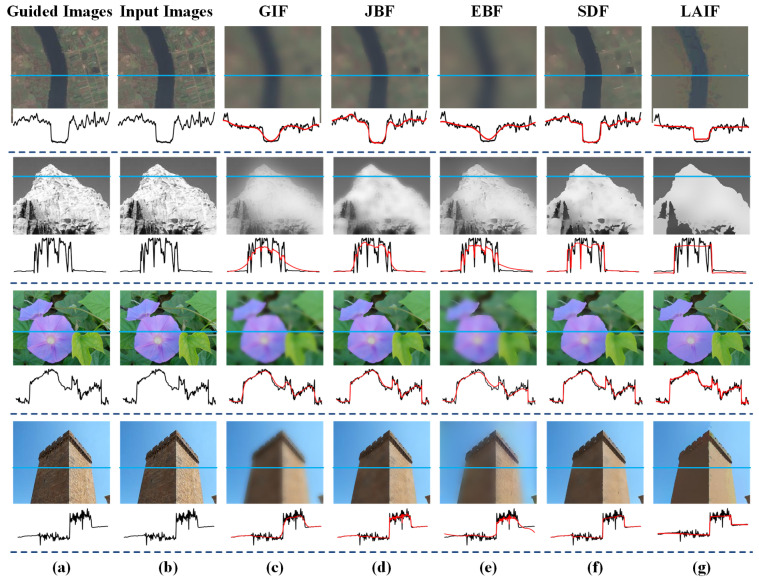
Filtering results via LAIF and other state-of-the-art methods, and the 1D detail illustration on the blue scanline (black corresponds to the input image, red corresponds to the output image). (**a**) Guided images. (**b**) Input images. (**c**) GIF. (**d**) JBF. (**e**) EBF. (**f**) SDF. (**g**) LAIF.

**Table 1 sensors-23-05776-t001:** Comparative analysis of methods (EBF, GIF, GBF, SDF, ours) on two public haze datasets (I-HAZE and O-HAZE) using the SSIM and PSNR metrics. The top two performing methods are underlined.

Dataset	Metric	Methods				
		EBF	GIF	GBF	SDF	Ours
I-HAZE	SSIM	0.7297	0.7205	0.7248	0.6217	0.7274
	PSNR	28.0868	28.0389	28.0834	27.8027	28.1024
O-HAZE	SSIM	0.7002	0.6891	0.6887	0.6117	0.7124
	PSNR	28.1271	28.1691	28.2235	27.8548	28.2481

**Table 2 sensors-23-05776-t002:** Flash/no-flash denoising and standard denoising capability evaluation based on SSIM and PSNR metrics. ‘w/o’ means without.

Metric	w/o Denoising	Flash/No-Flash Denoising					Standard Denoising			
		EBF	GIF	GBF	SDF	Ours	GF	MF	TV	Ours
SSIM	0.4022	0.7241	0.7105	0.7346	0.7312	0.7588	0.6978	0.6325	0.5568	0.7018
PSNR	28.0484	27.7332	28.2232	27.4847	27.5689	28.5654	27.6523	28.0335	27.8402	27.9041

**Table 3 sensors-23-05776-t003:** Comparative analysis of methods (EBF, GIF, GBF, SDF, ours) on three datasets (gblur, jp2k, jpeg) of LIVE Release2 using the SSIM and PSNR metrics. The top two performing methods are underlined.

Dataset	Metric	Methods				
		EBF	GIF	GBF	SDF	Ours
gblur	SSIM	0.7654	0.7609	0.6724	0.7394	0.7945
	PSNR	33.7678	29.1134	29.3686	30.0151	31.1163
jp2k	SSIM	0.6912	0.7048	0.5796	0.6283	0.7267
	PSNR	29.4225	28.9668	29.0575	29.4278	29.541
jpeg	SSIM	0.6534	0.6718	0.5238	0.5694	0.6912
	PSNR	28.9976	28.7603	28.7542	29.0288	28.8364

## Data Availability

Due to privacy concerns, it is inconvenient to provide.

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
