# Peer review of "Local Adaptive Image Filtering Based on Recursive Dilation Segmentation"

_sensors, 2023, doi:10.3390/s23135776_

Round 1

Reviewer 1 Report

This paper proposed a segmentation-based method of image filtering with applications to image dehazing, flash/no-flash denoising, detail enhancement and edge-preserving smoothing. Visual results have shown the effectiveness of the proposed method.

However, I have serval concerns on the paper.

1. The proposed framework is two-stage: segmentation first and then guided filtering, which is likely to suffer from the error of the first stage. Since the segmentation is done iteratively, why not make the two process jointly conducted, e.g., iteration between segmentation and filtering. In such a case, the error from the segmentation man be alleviated. 

2. There are no quantitative results for comparison, which is hard to justify the real performance of the proposed method, as images of visual results can be selected in bias.

3. The experiments of denoising are somehow rare. Since there are image pairs available, why not using the flash image directly for guidance? Further, standard denoising which is very suitable for testing structure-adaptive filtering methods should be done for improving the experiments. I encourage to follow the setting of [A] and compare with it (or at least have a discussion on it.

[A] Self2self with dropout: Learning self-supervised denoising from single image. CVPR 2020. 

4. Missing related works on nonlocal image filtering/smoothing, e.g.:

- Structure-texture image decomposition using discriminative patch recurrence. TIP 2021.

Missing references on deep image smoothing

A discussion on the relation to these works should be included.

The writing quality is quite good. A double check on typos and grammar errors would be better. 

Reviewer 2 Report

Evaluation is very poor

different filters image side by side is not good way to show the comparison.

Provide how edge preserving calculation is done, show the comparison of full data set of images edge preserving difference from original with other filters.

Round 2

Reviewer 1 Report

The revision has addressed my comments well.

Reviewer 2 Report

Satisfied with the authors response. Recommend for submission

Note: Increase figure image quality (1,3,4), and use a abbreviation table at the begining .